# Primary Tumor Resection in Synchronous Metastatic Colorectal Cancer Patients Treated with Upfront Chemotherapy plus Bevacizumab: A Pooled Analysis of TRIBE and TRIBE2 Studies

**DOI:** 10.3390/cancers15225451

**Published:** 2023-11-16

**Authors:** Valentina Fanotto, Daniele Rossini, Mariaelena Casagrande, Francesca Bergamo, Andrea Spagnoletti, Daniele Santini, Carlotta Antoniotti, Samanta Cupini, Francesca Daniel, Vincenzo Nasca, Guglielmo Vetere, Alberto Zaniboni, Beatrice Borelli, Martina Carullo, Veronica Conca, Alessandro Passardi, Emiliano Tamburini, Gianluca Masi, Nicoletta Pella, Chiara Cremolini

**Affiliations:** 1Department of Oncology, Academic Hospital of Udine, Azienda Sanitaria Universitaria Friuli Centrale (ASUFC), 33100 Udine, Italy; valentina.fanotto@asufc.sanita.fvg.it (V.F.); mariaelena.casagrande@asufc.sanita.fvg.it (M.C.); nicoletta.pella@asufc.sanita.fvg.it (N.P.); 2Department of Translational Research and New Technologies in Medicine and Surgery, University of Pisa, Via Savi 10, 56126 Pisa, Italy; daniele.rossini@unifi.it (D.R.); carlotta.antoniotti@unipi.it (C.A.); g.vetere1@studenti.unipi.it (G.V.); beatrice.borelli@phd.unipi.it (B.B.); m.carullo@studenti.unipi.it (M.C.); veronica.conca@phd.unipi.it (V.C.); gianluca.masi@unipi.it (G.M.); 3UO Oncologia 2 Universitaria, Azienda Ospedaliero-Universitaria Pisana, Via Roma 67, 56126 Pisa, Italy; 4Medical Oncology 1, Veneto Institute of Oncology IOV—IRCCS, 35128 Padua, Italy; francesca.bergamo@iov.veneto.it (F.B.); francesca.daniel@iov.veneto.it (F.D.); 5Department of Medical Oncology, Fondazione IRCCS Istituto Nazionale Dei Tumori, 20133 Milan, Italy; andrea.spagnoletti@unimi.it (A.S.); vincenzo.nasca@unimi.it (V.N.); 6Medical Oncology Unit A, Policlinico Umberto I, Sapienza University of Rome, 00161 Rome, Italy; daniele.santini@uniroma1.it; 7Department of Oncology, Division of Medical Oncology, Azienda USL Toscana Nord Ovest, 57124 Livorno, Italy; samanta.cupini@uslnordovest.toscana.it; 8Oncology Department, Istituto Ospedaliero Fondazione Poliambulanza, 25124 Brescia, Italy; alberto.zaniboni@poliambulanza.it; 9Department of Medical Oncology, IRCCS Istituto Romagnolo per lo Studio dei Tumori (IRST) “Dino Amadori”, 47014 Meldola, Italy; alessandro.passardi@irst.emr.it; 10Oncology Department and Palliative Care, Cardinale Panico Tricase City Hospital, 73039 Tricase, Italy; etamburini@piafondazionepanico.it

**Keywords:** metastatic colorectal cancer, primary tumor resection, prognosis, adverse events, first-line therapy, antiangiogenic drug

## Abstract

**Simple Summary:**

The management of primary tumors in metastatic colorectal cancer patients is still a broad and controversial scenario. While in the case of symptomatic primary tumors, the indication for surgery is a need rather than a choice, in the case of asymptomatic patients, literature data are conflicting about the benefit of primary tumor resection in terms of survival. This pooled analysis of patients enrolled in TRIBE and TRIBE2 studies revealed that primary tumor resection at baseline was independently associated with good prognosis and with lower incidence of serious gastrointestinal and surgical adverse events during upfront chemotherapy plus bevacizumab. Moreover, we observed that the benefit and toxicity profile of FOLFOXIRI plus bevacizumab was independent of the primary tumor resection. In the absence of strong evidence from randomized trials and considering the failure of many studies in this field, our results support the choice of primary tumor resection in selected asymptomatic patients.

**Abstract:**

Background: The decision to resect or not the primary tumor in asymptomatic patients with synchronous metastatic colorectal cancer (mCRC) is a complex and challenging issue for oncologists, especially when an antiangiogenic-based therapy is planned. Methods: Patients enrolled in the phase III TRIBE and TRIBE2 studies that compared upfront FOLFOXIRI + bevacizumab to FOLFIRI or FOLFOX + bevacizumab, respectively, were included. We assessed the association of primary tumor resection (PTR) with progression-free survival (PFS), overall survival (OS), response rate (ORR), rate of grade > 2 adverse events (AEs), and serious gastrointestinal and surgical AEs in the overall population and according to the treatment arm. Results: Of the 999 patients included, 513 (51%) underwent PTR at baseline. Longer PFS and OS were observed in resected patients compared to those with unresected primary tumors: 11.2 vs. 10.0 months (*p* < 0.001) and 26.6 vs. 22.5 (*p* < 0.001), respectively. In multivariate models, PTR was confirmed as an independent prognostic factor for better PFS (*p* = 0.032) and OS (*p* = 0.018). Patients with PTR experienced a higher incidence of grade 3 or 4 diarrhea (*p* = 0.055) and lower incidence of anemia (*p* = 0.053), perforation (*p* = 0.015), and serious gastrointestinal and surgical AEs (*p* < 0.001). No statistically significant differences were noted in incidence of bleeding (*p* = 0.39). The benefit of FOLFOXIRI + bevacizumab in terms of PFS (*p* for interaction: 0.46), OS (*p* for interaction: 0.80), ORR (*p* for interaction: 0.36), and incidence of grade 3 or 4 AEs was independent of PTR. Conclusions: PTR at baseline was independently associated with good prognosis in synchronous mCRC patients and with lower incidence of serious gastrointestinal and surgical AEs during upfront chemotherapy plus bevacizumab. The benefit and toxicity profile of FOLFOXIRI plus bevacizumab was independent of PTR.

## 1. Introduction

Around 25% of colorectal cancer (CRC) patients present with distant metastases at the time of first diagnosis, and about 50% of CRC patients will develop metastases after a curative surgery, mostly within the first five years. The surgical resection of radically resectable metastases (especially those located in the liver) is a potentially curative treatment with reported 5-year survival rates of 20–45%. In addition, initially unresectable metastatic CRC patients may be reconsidered for surgery in the case of volumetric response to systemic therapies with a clinically significant prognostic impact. The management of asymptomatic primary tumors in patients with synchronous metastases still remains a debated topic, especially when the use of treatments, including antiangiogenic drugs, is planned [1].

Upfront primary tumor resection (PTR) may avoid the occurrence of potentially life-threatening complications, including obstruction, bleeding, and perforation, that may compromise the administration of chemotherapy with or without antiangiogenic drugs [2] besides affecting patients’ quality of life (QoL). Moreover, elective surgery is associated with lower operative mortality than emergency procedures. Furthermore, PTR reduces the systemic burden of disease and may be associated with the reversal of systemic inflammation. Data suggest that the reversal of an elevated neutrophil/lymphocyte ratio after surgery is associated with better prognosis [3]. Conversely, a survival benefit from upfront PTR has never been demonstrated in prospective trials. Elective surgery may be associated with postoperative morbidity and perioperative mortality [4,5], and it certainly delays the administration of systemic therapy. At the same time, it should be considered that modern systemic therapies allow higher rates of early tumor shrinkage of both metastases and the primary tumor to be achieved, thereby potentially decreasing the possibility of primary-tumor-related complications. Furthermore, both preclinical and clinical data suggest that PTR may be associated with higher rates of systemic cancer spread and growth of pre-existing metastases, probably due to the shedding of circulating tumor cells and surgery-related immunosuppression [6]. Finally, limited evidence suggests that PTR may stimulate the angiogenesis of distant metastases, hypothesizing an antiangiogenic effect of the primary tumor [7,8,9]. 

Major guidelines currently recommend PTR in metastatic CRC patients only in the presence of overt symptoms or in the case of high risk of their imminent onset, while systemic treatment is recommended as the preferred initial step for asymptomatic patients [1,10,11], though it must be acknowledged that the management of the primary tumor is still an open issue [12]. Here, we present the results of a pooled analysis of two prospective, open-label, multicentric phase III randomized trials, TRIBE (NCT00719797) and TRIBE2 (NCT02339116), where untreated metastatic CRC patients received upfront chemotherapy (FOLFOXIRI, FOLFOX, or FOLFIRI) plus bevacizumab (bev). We aimed to assess the safety profile of study treatments according to PTR and the prognosis of enrolled patients.

## 2. Materials and Methods

This pooled analysis included synchronous metastatic CRC patients enrolled in the phase III TRIBE and TRIBE2 studies, which included patients aged 18–70 years with Eastern Cooperative Oncology Group performance status (ECOG PS) < 2 and patients aged 71–75 years with ECOG PS = 0. In the TRIBE study, 508 patients were randomized 1:1 to receive FOLFIRI/bev or FOLFOXIRI/bev for up to 12 cycles of induction chemotherapy, both followed by maintenance with 5-fluorouracil/bev until disease progression, unacceptable toxicities, or consent withdrawal. In the TRIBE2 study, 679 patients were randomly assigned to receive FOLFOX/bev (arm A) or FOLFOXIRI/bev (arm B) for up to 8 cycles of induction chemotherapy, both followed by maintenance with 5-fluorouracil/bev; after first disease progression, arm A received FOLFIRI/bev, whereas arm B received FOLFOXIRI/bev, both followed by the same maintenance until second disease progression, unacceptable toxicities, or consent withdrawal. In both studies, PTR was not a stratification criterion [13,14]. The primary objective of our study was to evaluate the impact of PTR in terms of toxicity of first-line chemotherapy plus bev, both in the overall study population and according to the treatment arm (triplet/bev vs. doublets (FOLFOX/FOLFIRI)/bev). Secondary objectives were to evaluate the prognostic impact of PTR at baseline in terms of progression-free survival (PFS) and overall survival (OS). All the analyses were conducted in the safety population, including all randomized synchronous metastatic CRC patients who received at least one cycle of treatment according to the randomization arm. Adverse events were graded according to the National Cancer Institute Common Terminology Criteria for Adverse Events (CTCAE) version 3.0 for the TRIBE trial and version 4.0 for the TRIBE2 study. Association between PTR and adverse events experienced during first-line therapy was analyzed by the χ^2^ test or Fisher’s exact test as statistically appropriate. Survival curves were estimated with Kaplan–Meier method and compared by log-rank test. Subgroup analyses of resected patients versus unresected patients for the occurrence of adverse events as well as subgroup analyses of doublets/bev versus FOLFOXIRI/bev for the occurrence of adverse events according to whether the primary tumor was resected were carried out using interaction tests.

## 3. Results

### 3.1. Patients

Overall, 999 of the 1187 randomized patients in the two trials were included in the safety population: 498 patients received FOLFOXIRI plus bev, while 501 patients receiving doublets (FOLFIRI or FOLFOX) plus bev. 513 (51%) underwent PTR before starting first-line therapy. Baseline characteristics of patients included in this analysis are described in Table 1. Compared with patients with unresected primary tumors, those who underwent PTR at baseline more frequently had right-sided colon cancer (*p* < 0.001), one metastatic site at the time of diagnosis of stage IV disease (*p* < 0.001), a liver-limited disease (*p* < 0.001), and a *BRAF*-mutated tumor (*p* = 0.007); moreover, they were more often women (*p* = 0.0087). In both groups, 50% of patients received FOLFOXIRI plus bev and the other 50% received a doublet plus bev.

A total of 206 (42%) of the 486 patients with unresected primary tumor at baseline underwent PTR at a later stage. These patients were younger (*p* = 0.0031) and more frequently had ECOG PS 0 (*p* = 0.0022), a liver-limited disease (*p* < 0.001), and one metastatic site (*p* < 0.001) compared to those with unresected primary tumors (Table 2).

### 3.2. Safety

Grade 3 or 4 adverse events occurring during first-line therapy are summarized in Figure 1. Among patients with unresected primary tumors, anemia (2.7% vs. 1.0%, *p* = 0.053), perforation (2.5% vs. 0.4%, *p* = 0.015), serious gastrointestinal adverse events (11.5% vs. 5.3%, *p* < 0.001), and serious surgical adverse events (9.5% vs. 2.3%, *p* < 0.001) were more frequent, while diarrhea occurred less frequently (10.9% vs. 15.0%, *p* = 0.055). In particular, patients with unresected primary tumors had more than double the probability of developing grade 3–4 anemia and serious gastrointestinal adverse events, more than six times the probability of experiencing perforation, and more than four times the probability of undergoing a serious surgical adverse event. No statistically significant differences were observed in the incidence of bleeding (0.8% vs. 0.4%, *p* = 0.39).

Of note, the probability of developing grade 3 or 4 adverse events based on whether the primary tumor was resected or not did not vary according to the treatment received in the first line (*p* > 0.05), as displayed in Figure 2.

### 3.3. Survival

At a median follow-up of 40.8 months (IQR 34.4–47.4 months), longer PFS and OS were observed in patients who underwent PTR compared to those with unresected primary tumor at the beginning of first-line therapy (median PFS: 11.2 vs. 10.0 months, hazard ratio (HR): 0.80, 95% confidence interval (CI): 0.70–0.91, *p* < 0.001; median OS: 26.6 vs. 22.5 months, HR: 0.78, 95% CI: 0.67–0.90, *p* < 0.001) (Figure 3). 

Triplet chemotherapy, ECOG PS score of 0, liver-only metastatic disease, one metastatic site, resected primary tumor at the beginning of first-line therapy, and *RAS/BRAF* wild-type tumors were identified as favorable prognostic factor for PFS and OS in the univariate analyses (Appendix A). In the multivariate analysis, PTR was confirmed as an independent prognostic factor for better PFS (*p* = 0.032), together with triplet chemotherapy (*p* = 0.001), ECOG PS score of 0 (*p* < 0.001), liver-only metastatic disease (*p* = 0.018), involvement of one metastatic site (*p* < 0.001), and *RAS/BRAF* wild-type status (*p* < 0.001) (Appendix A). PTR was also an independent prognostic factor for OS (*p* = 0.018), together with ECOG PS score of 0 (*p* < 0.001), involvement of one metastatic site (*p* = 0.0061), left-sided tumor (*p* = 0.018), and *RAS/BRAF* wild-type status (*p* < 0.001) (Appendix A).

Better PFS was observed in patients treated with FOLFOXIRI plus bev, both among those who underwent PTR before starting first-line therapy (12.9 vs. 9.9 months; HR: 0.70, 95% CI: 0.59–0.85) and among those with unresected primary tumors (11.3 vs. 9.3 months; HR: 0.78, 95% CI: 0.64–0.93) (*p* for interaction: 0.46). The benefit of triplet chemotherapy plus bev was also confirmed in terms of OS independently of PTR: 29.9 vs. 24.9 months in resected patients (HR: 0.82, 95% CI: 0.66–1.01) and 23.9 vs. 21.1 months (HR: 0.85, 95% CI: 0.69–1.04) (*p* for interaction: 0.80) (Figure 4). Similarly, FOLFOXIRI plus bev was associated with improved response in both subgroups: OR 1.28 (95% CI: 0.90–1.83) in resected patients and OR 1.62 (95% CI: 1.13–2.32) in unresected patients (*p* for interaction: 0.36).

## 4. Discussion

Our analysis fits into the broad and controversial scenario of the management of primary tumor in synchronous metastatic CRC patients. While in the case of symptomatic primary tumors, the indication for surgery is a need rather than a choice, in the case of asymptomatic primary tumors, literature data are conflicting about the benefit of PTR in terms of survival [12,15,16,17,18,19,20], as demonstrated by two recent meta-analyses mainly including nonrandomized, single-center, retrospective studies that came to opposite conclusions [19,20]. Notably, an intrinsic high risk of selection bias affected these analyses, together with the lack of data about QoL, administered treatments, reason for resection/no resection, baseline disease status, biological profile, and prognostic factor related to each individual patient. Some randomized controlled trials have been designed to address the topic of PTR in patients with unresectable stage IV CRC [12]. Some of these studies, such as ISAAC (NCT01086618) and SUPER (ACTRN12609000680268) were prematurely closed due to slow accrual. All the ongoing trials focus on asymptomatic patients, and in most of them, distant metastases should be judged unresectable by a multidisciplinary team. Recently, the results of two randomized, controlled phase III trials on this topic were released: iPACS and CAIRO4 trials [21,22]. The aim of the iPACS study was to demonstrate the superiority of PTR plus chemotherapy vs. chemotherapy alone in asymptomatic CRC patients with synchronous and unresectable metastases. The trial was prematurely discontinued due to futility as no difference in outcome between the two arms was observed after 22 months of median follow-up. The lack of clinically relevant information such as postprogression treatments, molecular markers, and QoL data should be taken into consideration when interpreting the trial’s results as well as, again, the slow accrual, with only 20% of the initially planned patients enrolled in seven years [21]. More recently, the CAIRO4 trial showed no significant OS difference among 206 patients with synchronous metastatic CRC amenable to palliative systemic therapy without severe symptoms related to the primary tumor according to PTR [22]. Previously, CAIRO4 investigators had published preliminary safety results showing higher 60-day mortality among patients randomized to PTR followed by systemic treatment (11% vs. 3%, *p* = 0.03) [23]. The target accrual of this study was also reduced, leading to a consequent decrease in the study power, from the 306 patients initially planned due to slow enrollment. Overall, the global low success of these trials confirms that several prognostic considerations weigh on the risk/benefit balance of PTR in the pragmatic evaluation of each case on an individual basis and that higher level of evidence to support clinical decisions in this field is hardly achievable. Surely, a key point is selection, and the efforts of the scientific community should therefore focus on identifying possible prognostic factors to select the best candidates for surgery.

In our retrospective analysis of two randomized trials, we found that patients who underwent PTR before starting their first-line therapy experienced better PFS and OS compared to those with unresected primary. A clear limitation is the lack of available information on the reason for resection/no resection before starting first-line therapy. Indeed, indications for PTR before enrollment in TRIBE and TRIBE2 studies were not collected, thus leading to inclusion in the resection group of both patients who had primary-related symptoms at diagnosis and patients who did not. Similarly, there was no information on the reason for non-resection, although the most likely hypothesis is the high burden and the apparent aggressiveness of the disease, leading to prioritization of systemic treatment. Actually, patients in the unresected group less frequently had a liver-only metastatic disease and/or a single anatomic site involved at diagnosis, thus not allowing a secondary radical resection of metastases to be foreseen as an achievable goal. Notably, the prognostic impact of PTR was retained in the multivariable model, including all prognostic features, both in terms of PFS and OS.

A strong point of our analysis is that patients received anticancer regimens that are nowadays widely used in clinical practice in contrast with outdated schedules adopted in other previous series, where the role of surgery might have been overestimated. Moreover, the combination with bev allows some useful information to be drawn on the role of PTR when antiangiogenic-based therapies are used. As expected, lower incidence of anemia, perforation, serious gastrointestinal adverse events, and serious surgical adverse events were reported in resected patients, although the absolute percentages of these events are quite low. No difference was found in the occurrence of bleeding. The impact of unresected primary tumor on bev-related toxicities was independent of the intensity of the chemotherapy backbone, i.e., it was not exacerbated or mitigated by the use of the triplet instead of conventional doublets. On the contrary, a higher incidence of diarrhea was observed in the resected group compared to patients with an intact primary, plausibly as a consequence of surgical resection.

## 5. Conclusions

In this pooled analysis of TRIBE and TRIBE2 studies, PTR at baseline was independently associated with good prognosis in synchronous metastatic CRC patients and with lower incidence of serious gastrointestinal and surgical adverse events during upfront chemotherapy plus bev. In the absence of strong evidence from randomized trials and considering the failure of many studies in this field, our results support the choice of PTR in synchronous metastatic CRC patients with no risk of immediate metastases-related symptoms and low disease burden. This may improve treatment tolerance, especially in the case of antiangiogenic-based regimens, thus reducing the risk of acute complications and serious gastrointestinal adverse events.

## Figures and Tables

**Figure 1 cancers-15-05451-f001:**
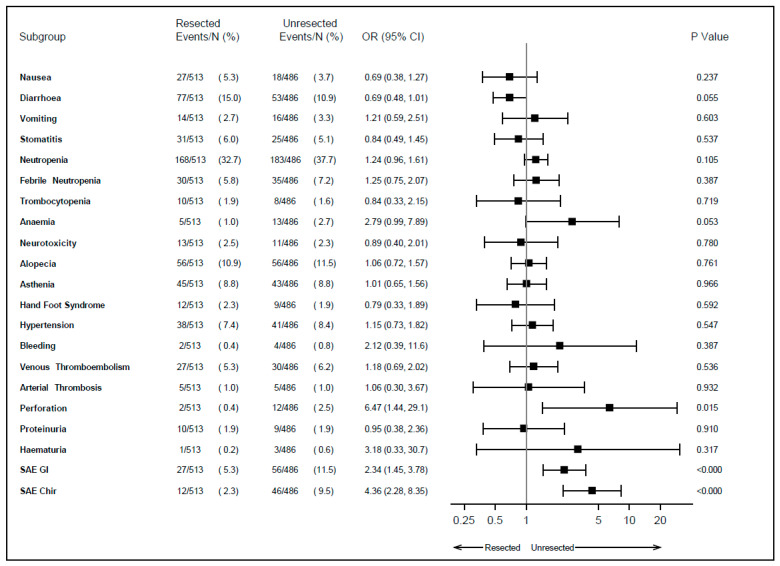
Adverse events during first-line therapy according to primary tumor resection. SAE: serious adverse event; GI: gastrointestinal; Chir: surgical; OR: odds ratio; CI: confidence interval.

**Figure 2 cancers-15-05451-f002:**
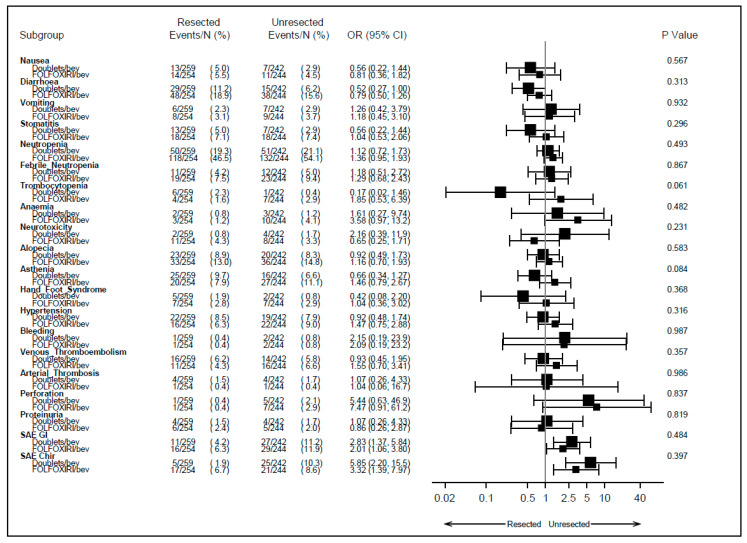
Adverse events according to primary tumor resection and treatment arm received in first line. SAE: serious adverse event; GI: gastrointestinal; Chir: surgical; OR: odds ratio; CI: confidence interval; bev: bevacizumab.

**Figure 3 cancers-15-05451-f003:**
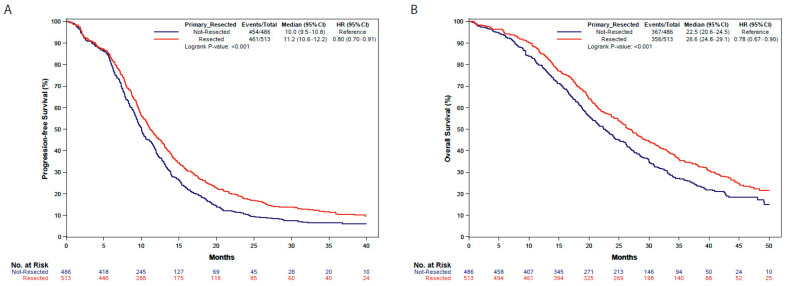
Progression-free survival (**A**) and overall survival (**B**) in resected and unresected patients. HR: hazard ratio; CI: confidence interval.

**Figure 4 cancers-15-05451-f004:**
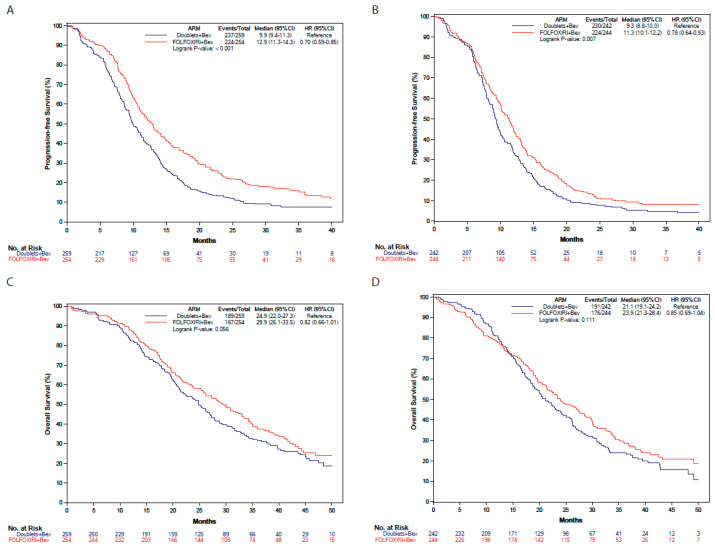
Progression-free survival and overall survival according to primary tumor resection and treatment arm. (**A**) Progression-free survival in resected patients according to the treatment arm. (**B**) Progression-free survival in unresected patients according to the treatment arm. (**C**) Overall survival in resected patients according to the treatment arm. (**D**) Overall survival in unresected patients according to the treatment arm. Blue: doublets plus bevacizumab; Red: FOLFOXIRI plus bevacizumab. HR: hazard ratio; CI: confidence interval.

**Table 1 cancers-15-05451-t001:** Baseline characteristics.

Baseline Characteristics	Resected Primary Tumor			
	No (n = 486)	Yes (n = 513)	Total (n = 999)	*p*-Value
Age (years)				0.7309 ^1^
N	486	513	999	
Mean (SD)	59.0 (9.4)	58.7 (9.9)	58.8 (9.7)	
Median	60.0	60.0	60.0	
Range	33.0, 75.0	29.0, 75.0	29.0, 75.0	
Age, n (%)				0.9412 ^2^
<70 years	417 (85.8%)	441 (86.0%)	858 (85.9%)	
≥70 years	69 (14.2%)	72 (14.0%)	141 (14.1%)	
Treatment arm, n (%)				0.8267 ^2^
Doublets CT + bevacizumab	242 (49.8%)	259 (50.5%)	501 (50.2%)	
FOLFOXIRI + bevacizumab	244 (50.2%)	254 (49.5%)	498 (49.8%)	
Gender, n (%)				0.0087 ^2^
Female	181 (37.2%)	233 (45.4%)	414 (41.4%)	
Male	305 (62.8%)	280 (54.6%)	585 (58.6%)	
ECOG PS, n (%)				0.0821 ^2^
0	413 (85.0%)	455 (88.7%)	868 (86.9%)	
1–2	73 (15.0%)	58 (11.3%)	131 (13.1%)	
Site of primary tumor, n (%)				<0.001 ^2^
Left rectum	325 (69.4%)	292 (57.4%)	617 (63.2%)	
Right rectum	143 (30.6%)	217 (42.6%)	360 (36.8%)	
Missing	18	4	22	
*RAS/BRAF* mutational status, n (%)				0.007 ^2^
*BRAF* mutated	25 (6.4%)	59 (12.8%)	84 (9.8%)	
*RAS* mutated	275 (70.0%)	301 (65.4%)	576 (67.5%)	
*RAS/BRAF* wild type	93 (23.7%)	100 (21.7%)	193 (22.6%)	
Missing	93	53	146	
Number of metastatic sites, n (%)				<0.001 ^2^
1	136 (28.1%)	216 (42.1%)	352 (35.3%)	
>1	348 (71.9%)	297 (57.9%)	645 (64.7%)	
Missing	2	0	2	
Liver-only disease, n (%)				<0.001 ^2^
No	379 (78.3%)	342 (66.7%)	721 (72.3%)	
Yes	105 (21.7%)	171 (33.3%)	276 (27.7%)	
Missing	2	0	2	

^1^ Kruskal–Wallis *p*-value; ^2^ chi-square *p*-value. CT: chemotherapy; ECOG PS: Eastern Cooperative Oncology Group Performance Status.

**Table 2 cancers-15-05451-t002:** Characteristics of patients who received subsequent primary tumor resection.

Characteristic	Subsequent PTR			
	No (n = 280)	Yes (n = 206)	Total (n = 486)	*p*-Value
Age (years)				0.0043 ^1^
N	280	206	486	
Mean (SD)	59.9 (9.7)	57.8 (8.9)	59.0 (9.4)	
Median	62.0	58.0	60.0	
Range	34.0, 75.0	33.0, 75.0	33.0, 75.0	
Age, n (%)				0.0031 ^2^
<70 years	229 (81.8%)	188 (91.3%)	417 (85.8%)	
≥70 years	51 (18.2%)	18 (8.7%)	69 (14.2%)	
Treatment arm, n (%)				0.1154 ^2^
Doublets CT + bevacizumab	148 (52.9%)	94 (45.6%)	242 (49.8%)	
FOLFOXIRI + bevacizumab	132 (47.1%)	112 (54.4%)	244 (50.2%)	
Gender, n (%)				0.9576 ^2^
Female	104 (37.1%)	77 (37.4%)	181 (37.2%)	
Male	176 (62.9%)	129 (62.6%)	305 (62.8%)	
ECOG PS, n (%)				0.0022 ^2^
0	226 (80.7%)	187 (90.8%)	413 (85.0%)	
1–2	54 (19.3%)	19 (9.2%)	74 (15.0%)	
Site of primary tumor, n (%)				0.0880 ^2^
Left rectum	177 (66.3%)	148 (73.6%)	325 (69.4%)	
Right rectum	90 (33.7%)	53 (26.4%)	143 (30.6%)	
Missing	13	5	18	
*RAS/BRAF* mutational status, n (%)				0.4153 ^2^
*BRAF* mutated	16 (7.1%)	9 (5.3%)	25 (6.4%)	
*RAS* mutated	160 (71.4%)	115 (68.0%)	275 (70.0%)	
*RAS/BRAF* wild type	48 (21.4%)	45 (26.6%)	93 (23.7%)	
Missing	56	37	93	
Number of metastatic sites, n (%)				<0.001 ^2^
1	52 (18.6%)	84 (41.0%)	136 (28.1%)	
>1	227 (81.4%)	121 (59.0%)	348 (71.9%)	
Missing	1	1	2	
Liver-only disease, n (%)				<0.001 ^2^
No	243 (87.1%)	136 (66.3%)	379 (78.3%)	
Yes	36 (12.9%)	69 (33.7%)	105 (21.7%)	
Missing	1	1	2	

^1^ Kruskal–Wallis *p*-value; ^2^ chi-square *p*-value. CT: chemotherapy; ECOG PS: Eastern Cooperative Oncology Group Performance Status; PTR: primary tumor resection.

## Data Availability

The data presented in this study are available on request from the corresponding author.

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
