# Peer review of "Primary Tumor Resection in Synchronous Metastatic Colorectal Cancer Patients Treated with Upfront Chemotherapy plus Bevacizumab: A Pooled Analysis of TRIBE and TRIBE2 Studies"

_cancers, 2023, doi:10.3390/cancers15225451_

Round 1

Reviewer 1 Report

Comments and Suggestions for Authors

The authors address the question of whether primary tumor resection is beneficial in metastatic colorectal carcinoma. Patients with inoperable distant metastasis and upfront chemotherapy with bevacizumab were included (patients from the TRIBE and TRIBE2 studies).

This is an interesting question for patients with unresectable synchronous distant metastases. But not for patients with metachronous distant metastases. In these patients, primary tumour resection was performed in 167/176=95% of patients. It is also known that patients with metachronous distant metastases have a better prognosis than patients with synchronous distant metastases. These patients with metachrounous distant metastases should be excluded from the analysis because the question of primary tumor resection does not usually arise in metachronous distant metastasis. This should also have eliminated patients with ‘prior adjuvant therapy’, which also caused confusion.

Otherwise, the paper is carefully written and well discussed.

Author Response

We thank you very much for taking the time to review this manuscript. Following your suggestion, which in our opinion highlighted a very significant point, we restricted the analysis to synchronous metastatic CRC patients, in whom primary tumor resection represents a relevant problem. We therefore modified the manuscript and related figures and tables accordingly.

Reviewer 2 Report

Comments and Suggestions for Authors

This is a pooled analysis of TRIBE and TRIBE2 clinical trials that evaluated whether primary tumor resection (PTR) was associated with improved clinical outcomes in patients who enrolled in these two phase 3 studies. The findings are important; however, some important clarifications are needed. Please consider addressing the following comments. 

More than and about 50% of patients in the TRIBE and TRIBE2 studies underwent surgical resection of primary tumor, respectively, based on the primary publications of these two clinical trials. Given that both studies were designed to compare the efficacy of two systemic therapy regimens in patients with unresectable metastatic disease, the higher rates of PTR suggest that perhaps most of these patients underwent surgical resection of the primary tumor around the time of initial diagnosis of localized disease (ie, Stage I, II and III) instead of around the time of enrollment to the study.

If this is correct, it will be important to specify 1) the percent of patients who received PTR for localized disease vs those who received PTR despite diagnosis of stage IV disease, 2) the duration of time between PTR and enrollment to the TRIBE/TRIBE 2 clinical trials, 3) the intent of the surgery (ie, curative vs palliative), and 4) whether PTR was still associated with improved clinical outcomes after accounting for histologic grade of primary tumor, stage of cancer when PTR was performed, and the duration of time interval between PTR and enrollment to clinical trials. 

Based on the answers to the above questions, one may not be able to conclude that PTR in asymptomatic patients should be supported. For instance, patient A underwent PTR for stage I disease who developed recurrent cancer 5 years later and enrolled to study vs patient B presented with unresectable stage IV disease who did not undergo PTR and enrolled to study. Obviously, patient A would more likely to have a better clinical course than patient B but this difference is unlikely explained by PTR.  

Author Response

We thank you very much for taking the time to review this manuscript. We really appreciated your comments which pushed us to better define the study population. Given that patients with synchronous and metachronous metastatic disease represent two very different settings and that in clinical practice primary tumor resection represents a relevant point in synchronous metastatic CRC population, we decided to restrict the analysis to synchronous metastatic CRC patients. We therefore modified the manuscript and related figures and tables accordingly.

Round 2

Reviewer 1 Report

Comments and Suggestions for Authors

Well done.

Author Response

We thank you very much for the additional time spent reviewing our work. We are pleased to hear that you were satisfied with our revision.

Reviewer 2 Report

Comments and Suggestions for Authors

Thank you for performing additional analysis to address the comments. 

Please consider the following suggestions. 

1) Introduce the term "synchronous" to the title 

2) Revise the column heading, "PTR under treatment", in Table 2. I believe based on the text, No and Yes refers to whether patients received subsequent primary tumor resection.  

Comments on the Quality of English Language

Abbreviations (.e.g, bev) were not defined. 

Author Response

We thank you very much for the additional time spent reviewing our work. Following your suggestions, we have introduced the term "synchronous" to the title and we have modified the column heading in Table 2 from “PTR under treatment” to "Subsequent PTR”. We have defined all the abbreviations.